# p53 Transactivation Domain Mediates Binding and Phase Separation with Poly-PR/GR

**DOI:** 10.3390/ijms222111431

**Published:** 2021-10-22

**Authors:** Sinem Usluer, Emil Spreitzer, Benjamin Bourgeois, Tobias Madl

**Affiliations:** 1Gottfried Schatz Research Center for Cell Signaling, Metabolism and Aging, Department of Molecular Biology and Biochemistry, Medical University of Graz, 8010 Graz, Austria; sinem.usluer@medunigraz.at (S.U.); emil.spreitzer@medunigraz.at (E.S.); benjamin.bourgeois@medunigraz.at (B.B.); 2BioTechMed-Graz, 8010 Graz, Austria

**Keywords:** poly-PR/GR, neurodegenerative disease, LLPS, p53, intrinsically disordered domains, membraneless organelles

## Abstract

The most common genetic cause of amyotrophic lateral sclerosis (ALS) and frontotemporal dementia (FTD) is the presence of poly-PR/GR dipeptide repeats, which are encoded by the chromosome 9 open reading frame 72 (C9orf72) gene. Recently, it was shown that poly-PR/GR alters chromatin accessibility, which results in the stabilization and enhancement of transcriptional activity of the tumor suppressor p53 in several neurodegenerative disease models. A reduction in p53 protein levels protects against poly-PR and partially against poly-GR neurotoxicity in cells. Moreover, in model organisms, a reduction of p53 protein levels protects against neurotoxicity of poly-PR. Here, we aimed to study the detailed molecular mechanisms of how p53 contributes to poly-PR/GR-mediated neurodegeneration. Using a combination of biophysical techniques such as nuclear magnetic resonance (NMR) spectroscopy, fluorescence polarization, turbidity assays, and differential interference contrast (DIC) microscopy, we found that p53 physically interacts with poly-PR/GR and triggers liquid–liquid phase separation of p53. We identified the p53 transactivation domain 2 (TAD2) as the main binding site for PR25/GR25 and showed that binding of poly-PR/GR to p53 is mediated by a network of electrostatic and/or hydrophobic interactions. Our findings might help to understand the mechanistic role of p53 in poly-PR/GR-associated neurodegeneration.

## 1. Introduction

Amyotrophic lateral sclerosis (ALS) and frontotemporal dementia (FTD) are neurodegenerative diseases with many common neuropathological, genetic, and clinical features [1,2]. The most frequent genetic cause of ALS and FTD is a GGGGCC hexanucleotide repeat expansion in the first non-coding region of the chromosome 9 open reading frame 72 gene (C9orf72) [3,4]. This mutation results in loss of function of the C9orf72 by inhibition of transcription [3,5,6]; generation of RNA foci [7], which sequester critical RNA-binding proteins [8,9] and/or interfere with essential cellular processes [10]; and repeat-associated non-AUG (RAN) translation of dipeptide repeat proteins (DPRs) [11,12,13,14]. Five DPRs can be translated, including glycine-alanine (GA) and glycine-arginine (GR) repeats from the sense strand, proline-alanine (PA) and proline-arginine (PR) repeats from the anti-sense strand, and glycine-proline (GP) from both strands [12,13]. Accumulation of these DPRs in the brains of patients with C9orf72 mutants has been shown [15]. Among them, poly-PR and poly-GR are the most toxic ones. The addition of recombinant PR or GR polymers to HeLa or human astrocytes cells and expressing 50 repeats of GR or PR in *Drosophila* result in RNA processing alterations and early lethality, respectively [15,16,17,18,19,20,21]. Moreover, expression of poly(GR)_100_ or poly(PR)_28_ causes early-onset and severe neurodegeneration with motor dysfunction and memory impairment in mice [14,22,23]. Toxicity of poly-PR/GR has been shown to be linked to multiple cellular mechanisms, such as mitochondrial stress [24], ER stress [25,26], impaired nucleocytoplasmic transport [20,27,28], oxidative stress [24], impaired splicing [29,30,31], impaired translation [32], interference with the formation of membrane-less organelles (MLOs) [33,34,35,36], and combinations thereof.

DPRs undergo phase separation in the presence of a crowding agent in vitro [37]. Overexpression of poly-PR/GR in cells increases the formation of stress granules, whereas they disrupt nucleolus structure and function [35,36,37] which in turn is associated with p53 stabilization during DNA damage response [38]. Additionally, overexpression of poly-PR/GR DPRs in neuronal cells triggers DNA damage [39]. It remains, however, elusive whether poly-PR/GR-mediated DNA damage or disruption of nucleolus function stabilize p53 in neurodegenerative diseases and whether those mechanisms are mediated by direct interactions between p53 and poly-PR/GR. Recently, Maor-Nof et al. showed that poly(PR)_50_ expression in mouse primary cortical neurons increases the accessibility of binding sites for the transcription factor p53, which enables the activation of downstream p53 transcriptional regulators. A reduction in p53 protects against axonal degeneration, cell death, early lethality, DNA damage, and eye photoreceptor degeneration in primary cortical neurons, C9orf72 mouse/fly models, and induced pluripotent stem cells (iPSCs) derived from ALS patients, respectively [40]. However, the mechanistic details leading to poly-PR mediated stabilization and activation of p53 remain enigmatic.

Here, we showed using nuclear magnetic resonance (NMR) spectroscopy and fluorescence polarization (FP) that p53 directly interacts with poly-PR and poly-GR repeats, involving the p53 transactivation domain (TAD). Among the p53 TAD, we showed that only the TAD2 subdomain is sufficient to mediate the interaction with DPRs, but the TAD1 also contributes to the increase in the overall binding affinity. We showed that the interaction between poly(PR)_25_/poly(GR)_25_ and p53 TAD is mediated by a network of hydrophobic and electrostatic interactions. We then addressed the question of whether DPRs binding to p53 affects their propensity to phase separate. We showed using turbidity assay and differential interference contrast (DIC) microscopy that phase separation of p53^1−94^ and p53^1−312^, including TAD and DNA-binding domain (DBD), is mediated by poly(PR)_25_/poly(GR)_25_. Besides intermolecular interaction between p53 and poly(PR)_25_/poly(GR)_25_, phase separation of p53^1−312^ is enhanced by the intramolecular interaction between p53^1−94^ and p53^DBD^. Understanding of the interaction of p53 and poly(PR)_25_/poly(GR)_25_ and its effect on the phase separation in vitro helps to understand the toxicity of poly-PR/GR mechanistically, which might be used for therapeutic approaches.

## 2. Results

### 2.1. Poly(PR)_25_/Poly(GR)_25_ Interact Directly with p53

As previous studies have shown a link between DPRs, p53 activation, and stabilization [40], we wanted to assess whether the model DPRs poly(PR)_25_ and poly(GR)_25_ directly interact with p53 in vitro using the purified proteins and NMR spectroscopy.

To this end, we purified recombinant isotope-labelled p53^1−312^ (residues 1 to 312), which includes the N-terminal TAD; the poly-proline region; the DBD; and performed titrations with either synthetic poly(PR)_25_ or poly(GR)_25_ dipeptide repeats named PR25 and GR25, respectively. We excluded the C-terminal tetramerization domain that would prevent obtaining high-quality NMR data (Figure 1A). The addition of one stoichiometric equivalent of either unlabeled PR25 or GR25 caused disappearance and chemical shift perturbations (CSPs) of the p53^1−312 1^H-^15^N HSQC cross-peaks (Figure 1B,C), which indicates direct binding of PR25/GR25 to the corresponding residues of p53^1−312^ in the fast to intermediate NMR exchange time scale. Inspection of the peaks showed that, mostly, the intense peaks belonging to the disordered N-terminal region seemed to be affected by PR25/GR25.

In order to localize the binding site of PR25/GR25 on p53, we carried out a divide-and-conquer approach and purified the isolated N-terminal disordered region containing the transactivation domains 1 and 2 as well as the poly-proline region (p53^1−94^, residues 1 to 94) and a construct harboring the p53 DBD (p53^DBD^; residues 95 to 312). ^1^H-^15^N HSQC NMR spectra of isotope-labelled p53^1−94^ and p53^DBD^ were recorded in the absence and presence of one stoichiometric equivalent of PR25/GR25. The addition of PR25/GR25 to the p53^DBD^ did not change the ^1^H-^15^N HSQC NMR spectra of p53^DBD^ (Figure 1D,E), which indicates that this p53 domain is not involved in the interaction with PR25/GR25. In contrast, and in line with the observations for p53^1−312^, p53^1−94 1^H-^15^N HSQC cross-peaks showed CSPs in the presence of PR25/GR25 (Figure 1F,G), which indicates that the interaction between p53 and PR25/GR25 is mediated by residues localized within this N-terminal intrinsically disordered region. Inspection of the residues affected by the addition of PR25/GR25 showed that the strongest CSPs cluster within p53^TAD2^, for both PR25 and GR25 (Figure 1F,G). In summary, our NMR analysis shows that PR25/GR25 directly bind to p53, involving residues clustering in p53^TAD2^.

### 2.2. p53 Transactivation Domain 2 Is the Main PR25/GR25 Binding Site

To specify the role of p53^TAD1^ and p53^TAD2^ in PR25/GR25 binding, isotope-labelled p53^TAD1^ and p53^TAD2^ were purified separately, and ^1^H-^15^N HSQC NMR experiments were performed in presence or absence of PR25/GR25. p53^TAD1 1^H-^15^N HSQC cross-peaks showed no CSPs upon the addition of one stoichiometric equivalent of PR25/GR25 (Appendix A). In contrast, and in line with the results obtained for the longer constructs, p53^TAD2 1^H-^15^N HSQC cross-peaks showed CSPs upon PR25/GR25 addition (Figure 2A,B). This shows that TAD2, but not TAD1, is sufficient to mediate PR25/GR25 binding. Inspection of the TAD2 residues affected upon addition of PR25/GR25 showed that the C-terminal end (51–55) is mainly involved. Addition of PR25 or GR25 yielded similar CSPs.

To complement the NMR data, we performed FP experiments in order to determine binding affinities of p53^1−94^ and p53^TAD2^ for fluorescein isothiocyanate (FITC)-labeled PR25 and GR25 peptides (Figure 2C,D). We found that p53^1−94^ has similar low micro molar range affinity for PR25 and GR25 with equilibrium dissociation constants (K_d_s) of 1.84 ± 0.72 and 0.43 ± 0.17 µM for PR25 and GR25, respectively. Similarly, p53^TAD2^ has low micro molar range affinity for both PR25 and GR25 with K_d_s of 3.64 ± 1.16 and 2.85 ± 0.51 µM for PR25 and GR25 binding, respectively. Compared with p53^TAD2^, p53^1−94^ has approximately twofold/sixfold higher affinities for PR25 and GR25 binding, respectively (Figure 2C,D). Even though p53^TAD1^ is insufficient to bind PR25 and GR25 (Appendix A), it enhances binding to PR25/GR25 in the p53^1−94^ construct (Figure 2C,D). Overall, we observed that the binding affinities of PR25/GR25 for p53 are similar comparing the results obtained for p53^1−94^ and p53^TAD2^, with slightly enhanced affinity of GR25. Given the similar CSPs observed for both GR25 and PR25, and the absence of binding to isolated p53^TAD1^, we conclude that p53^TAD2^ is sufficient for PR25/GR25 binding.

### 2.3. Binding of PR25/GR25 to p53^1−94^ Increases Its Rigidity and α-Helical Secondary Structure Propensity

The TAD of p53 is intrinsically disordered, but has been shown previously to adopt secondary α-helical structure upon binding to different protein partners [41,42,43]. In order to obtain information on p53^1−94^ flexibility in the presence of DPRs, we recorded ^15^N{^1^H} heteronuclear NOE experiments of isotope-labeled p53^1−94^ in the presence and absence of PR25/GR25. ^15^N{^1^H} heteronuclear NOEs report the motion of individual N-H bond vectors and provide information about the rigidity of the protein backbone, in which positive values correspond to rigidity and negative values represent flexible disordered regions. We observed that HetNOEs increased in the presence of PR25/GR25, in the regions comprising TAD1 (from 17 to 37) and TAD2 (from 38 to 55), which indicates that binding of PR25/GR25 enhances the rigidity of p53 TAD1 and TAD2 (Figure 3A,B). However, there are few residues that exhibit higher flexibility when bound to PR25, namely Gln38 and 51Glu. In complex with GR25, residues 21Asp, 24Lys, and 56Glu become more flexible (Figure 3A,B).

To test if PR25/GR25 binding affects the secondary structure of p53^1−94^, CBCA(CO)NH NMR experiments of isotope-labeled p53^1−94^ were recorded in the absence and presence of PR25/GR25. From these experiments, we obtained the ^13^C chemical shifts of the alpha (α) and beta (β) carbon atoms of the p53 residues, which can be used to obtain information on the propensity of the corresponding residues to adopt either a α-helical or β-stranded secondary structure. The p53 TAD1 region (residues 17–28) harbors a stretch of positive Δδ(Cα-Cβ) values in the presence and absence of PR25/GR25, showing that this p53 domain is prone to form α-helical secondary structure on its own. This propensity remains unaltered upon addition of PR25/GR25, which is in line with the absence of binding to this region (Figure 3C,D). In contrast, we observed a slight increase in the Δδ(Cα-Cβ) values in the TAD2 region, especially for residues 48Asp, 49Asp, 50Ile, and 51Glu in the presence of PR25, as well as residue 48Asp in the presence of GR25, suggesting that this p53 domain harbors increased α-helical propensity upon DPR binding (Figure 3C,D). It is worth noting that binding of p53 to GR25 induces extensive line broadening; therefore, several of the ^13^C chemical shifts of the TAD2 could not be assigned. In summary, the interaction of PR25/GR25 with the p53 N-terminal disordered region increases the overall rigidity of the region encompassing the TAD1 and TAD2 sub-domains. While the TAD1 is already α-helical in the absence of DPRs and helical propensity is unchanged upon DPR binding, the TAD2 becomes α-helical only upon binding.

### 2.4. Interaction between PR25/GR25 and p53^1−94^ Is Mediated by Electrostatic and/or Hydrophobic Interactions

To obtain insight into the molecular details of how p53 binds PR25 or GR25, ^13^C and^15^N filtered and ^13^C-edited 3D nuclear Overhauser effect spectroscopy (NOESY) experiments were recorded with ^13^C-^15^N isotope-labelled p53^1−94^ in the presence of either PR25 or GR25. These types of experiments provide information about intermolecular interactions by eliminating NOEs from intramolecular interactions in each of the components of the complex [44]. We observed NOE cross peaks from each carbon-attached ^1^H position involving ambiguous p53^1−94^ Glu-Cγ, Glu/Gln-Cβ, Pro-Cγ, Leu-Cδ, Ala-Cβ, and Met-Cε for the GR25-bound complex, indicating that these p53 residues are in close proximity with GR25 protons (Figure 3E,G). In the same way, we observed NOE cross from each carbon-attached ^1^H position involving p53^1−94^ Leu-Cδ, Thr-Cγ, and Met-Cε for the PR25-bound complex (Figure 3F,H). Although the poor chemical shift dispersion in the ^1^H,^13^C HSQC spectrum of p53^1−94^ did not allow for residue-specific assignment of NOE cross peaks, we can, based on the CSP results, tentatively assign these residues to be in close proximity with residues clustered in p53 TAD2 such as Glu51/Glu56, Gln38/Gln52, Pro47, Leu43/Leu45, Ala39, Met40/Met44, and Thr55. Inspection of the NOESY cross peaks revealed that these p53 residues showed cross peaks to ^1^H chemical shifts characteristic for arginine, proline, and glycine residues. The nature of these interactions was mostly hydrophobic, as indicated by cross peaks between p53 leucines and methionines to either PR25 arginines or PR25 prolines. In a similar manner, the interaction between p53 and GR25 is mostly mediated via electrostatic interactions from p53 glutamates and arginines from GR25, but also hydrophobic contact between (i) p53 methionines and prolines to arginines of GR25; (ii) p53 leucines and prolines to either glycines or arginines of GR25; and (iii) p53 alanines to glycines of GR25.

In summary, we demonstrated that binding and interaction of p53 with poly-PR/GR are mediated mainly by p53 TAD2 and involve a network of electrostatic and/or hydrophobic interactions.

### 2.5. PR25/GR25 Induce Phase Separation of p53^1−94^ and p53^1−312^

It has been previously shown that poly-PR/GR can regulate phase separation of proteins with low complexity domains (LCDs) [45]. Furthermore, other studies have shown that p53 can phase separate *in vitro* and associate with MLOs in cells [46,47]. To understand whether PR25/GR25 affect phase separation of p53, we performed turbidity assays and DIC microscopy with different p53 constructs harboring the (i) N-terminal disordered region and the DBD (p53^1−312^), (ii) the N-terminal disordered region (p53^1−94^), (iii) TAD2 and (iv) DBD. Titration of increasing concentrations of PR25/GR25 with fixed (50 µM) concentrations of p53^1−312^ resulted in increasing turbidity at low PR25/GR25 concentrations (Figure 4A,B). In contrast, higher concentrations of PR25/GR25 (especially PR25) suppressed phase separation of p53^1−312^ (Figure 4A,B). Phase separation of fixed (50 µM) concentrations of p53^1−94^ reached a maximum at 150 µM GR25 and 100 µM PR25, respectively. Higher concentrations of PR25/GR25 suppressed LLPS (Figure 4A,B). Among the p53 constructs tested, p53^TAD2^ shows a slight propensity to phase separate only in the presence of 40 to 50 µM GR25, whereas its phase separation was not induced by PR25 (Figure 4A,B). As expected, addition of increasing concentration of PR25/GR25 with p53^DBD^ did not change the phase separation propensity of p53^DBD^ because there is no direct binding between PR25/GR25 and p53^DBD^.

We further validated our findings by monitoring the phase separation of p53^1−312^, p53^1−94^, and p53^TAD2^ by DIC microscope. To record the formation and fusion of droplets, different concentrations of PR25/GR25 were used for different constructs in order to observe the light phase and avoid the dense phase [48]. Droplet formation and their fusions were observed and recorded over time. In line with the turbidity assays, we observed droplet formation of p53 constructs except p53^DBD^ upon addition of PR25/GR25 (Figure 4C–E,G). There was no droplet formation for p53^TAD2^ in the presence of PR25 (Figure 4F). In summary, our turbidity and DIC results showed that poly-PR/GR regulate p53 LLPS via direct interactions.

## 3. Discussion

In this work, we found that p53 directly interacts with poly-PR/GR. We showed that the interaction between poly-PR/GR and p53 is mediated via the N-terminal TAD of p53 harboring two sub-domains, namely TAD1 (residues 1–39) and TAD2 (residues 40–61) [49] (Figure 1B–G, Figure 2A,B and Appendix A). We observed that p53^TAD2^ is sufficient to bind to PR25/GR25 and that the presence of other regions within the N-terminal TAD slightly enhanced the affinity towards PR25/GR25 (Figure 2C,D). The intrinsically disordered N-terminal p53 TAD is involved in several protein–protein interactions, and p53 TAD2 alone has been reported to bind to CBP/p300 (CREB (cAMP-response-element-binding protein)-binding protein), high mobility group B1 (HMGB1), replication protein A, transcription factor B1 subunits of human and yeast transcription factor II H complex, and metastasis-associated protein S100A4 [41,50,51,52,53,54,55]. S100A4 is overexpressed in a range of different tumor types [56]. Its oncogenic property involved inhibition of p53-dependent growth arrest and apoptosis through enhancing p53 degradation. The presence of poly-PR/GR might compete with the binding of S100A4, which might promote p53 function and stability [56]. CBP/p300 stabilizes p53 in response to DNA damage by preventing MDM2-mediated p53 degradation [57,58,59,60]. Binding of poly-PR/GR to p53 TAD2 might interfere with binding to CBP/p300 during DNA damage or under stress conditions and, in turn, might modulate p53 stability and function.

We demonstrated that binding of PR25/GR25 to p53^1−94^ enhances the rigidity and formation of α-helical propensity (Figure 3A–D). For both TADs, it has been reported that they show α-helical propensity in the absence of binding partners [61], and that binding to folded proteins, such as the nuclear coactivator binding domain of CBP, results in folding and increased helicity of p53 TAD2 [41]. We showed that interactions between p53^1−94^ and PR25/GR25 are mediated by a network of electrostatic and/or hydrophobic interactions (Figure 3E-H and Figure 5B). It is known that p53 TAD interactions with others are also mediated by both electrostatic and/or hydrophobic interactions [61]. Interactions between TAD2 and the DNA-binding HMG box are both hydrophobic and electrostatic, involving the minor-groove-binding residues and basic DNA-binding residues in the HMG box, respectively [62]. Thus, it is tempting to speculate that PR25/GR25 could compete with proteins binding to p53 TAD2, such as HMGB1, and modulate their function.

In addition to the important role of p53 in cancer, p53 stabilization and activation by C9orf72 poly-PR seem to be key mechanisms causing the neurodegenerative phenotype in primary cortical neurons expressing C9orf72 poly-PR, iPSCs from ALS patients with C9orf72 GGGGCC, transgenic fly expressing C9orf72 GGGGCC, and transgenic mouse expressing C9orf72 poly-PR [40]. Moreover, it has been shown that poly(GR)_80_ expression in iPSC-derived motor neurons increased DNA damage, which leads to a pro-apoptotic response by activating p53 [24,39,63]. However, it remained unclear whether a direct interaction between p53 and poly-PR/GR could explain the phenotype. Our findings help to understand how stabilization of p53 and activation of the p53 transcriptional pathway are regulated and how these interactions could be targeted in neurodegenerative disease related to C9orf72 dipeptide repeat proteins.

Both poly-PR/GR and p53 have been reported to undergo LLPS individually; however, whether poly-PR/GR can directly interact with p53 and thereby regulate p53 LLPS remained elusive. We showed that phase separation of p53 can be triggered by PR25/GR25, and that an interaction between p53^DBD^ and p53^TAD2^ contributes strongly to LLPS (Figure 4A,B,D and Appendix A). This suggests that LLPS of p53^1−312^ is driven by a network of intermolecular interactions between p53^TAD2^ and PR25/GR25 and intermolecular interactions between p53^TAD2^ and p53^DBD^ (Figure 5A). The dissociation of droplets observed at high concentrations of PR25/GR25 is likely due to saturation of the inter-molecular interactions between p53^TAD2^ and p53^DBD^. This model also implies that PR25/GR25 competition with the intra/inter-molecular p53^TAD2^–p53^DBD^ interaction could lead to an activation of p53, through loss of the previously described auto-inhibition [64,65,66], which is in line with the enhanced p53 transcriptional activity observed in poly-PR disease models [40].

Aberrant LLPS through poly-PR/GR has been reported for several disease-related RNA-binding proteins, including FUS, TDP-43, hnRNPA1, hnRNP2B1, TIA1, Ataxin-2, and Matrin-3 [33,45,67]. DPRs can alter the liquid-to-solid transition and aggregation of RNA-binding proteins and, in turn, MLO formation in cells [45]. Overexpression of poly-PR/GR, for example, leads to disruption of nucleolar phase dynamics by binding of poly-PR/GR to nucleolar components like nucleophosmin-1. Given that p53 stability is regulated in nucleoli and that functional nucleoli are required for MDM2-mediated p53 degradation [38,68], disruption of nucleoli by poly-PR/GR might explain the accumulation of p53 [36,38,45]. In the future, it will be interesting to investigate how p53 LLPS is involved in these (patho)physiological processes.

Poly-PR/GR mediated LLPS of p53 observed here might be of general importance in the regulation of transcriptional condensates. Activation domains from transcription factors, including p53, form phase-separated condensates with the mediator to activate gene expression [69]. In the future, it will be interesting to reveal if poly-PR/GR modulates the formation of p53 transcriptional condensates and, through this, regulates the expression of p53 target genes.

As a summary, we demonstrate that binding of p53 to PR25/GR25 is mediated by mainly p53^TAD2^ and that phase separation of p53 is regulated by PR25/GR25 in vitro. This might help to understand the mechanistic role of p53 in poly-PR/GR dipeptide repeat related diseases such as ALS and FTD.

## 4. Materials and Methods

### 4.1. Plasmids and Synthetic Peptides

Constructs for *Escherichia coli (E. coli)* expression of human p53 from amino acids 1 to 312 (p53^1−312^), amino acids 95 to 312 (p53^DBD^), amino acids 14 to 37 (p53 ^TAD1^), amino acids 37 to 57 (p53 ^TAD2^), and amino acids 1 to 94 (p53^1−94^) were generated by synthesis of the corresponding optimized p53 cDNA constructs from Genscript and insertion of these cDNA into pETM11-ZZ-His_6_ vector via NcoI/BamHI restriction digest.

Synthetic peptides were synthesized and HPLC-purified by Peptide Specialty Laboratories GmbH at Heidelberg, Germany. Peptides were obtained as lyophilized powder.

### 4.2. Protein Expression and Purification

To express the recombinant unlabeled or ^15^N labeled or ^15^N^13^C labeled ZZ-His_6_ proteins, different bacterial expression pETM11-ZZ-His_6_ vectors were transformed into *E. coli* BL21-DE3 Star strain. On the next day, a single colony was picked and grown overnight in 20 mL lysogeny broth (LB) medium supplemented with 50 mg/L kanamycin. Then, 10 mL of the pre-culture was transferred to 1 L LB or into minimal medium supplemented with either 6 g of unlabeled glucose or 2 g of ^13^C_6_H_12_O_6_ (Cambridge Isotope Laboratories, Saarbrücken, Germany) and 3 g of either unlabeled NH_4_Cl or ^15^NH_4_Cl (Merck, Darmstadt, Germany). When OD (600 nm) reached 0.8, 1 L of cells was induced with 0.5 mM Isopropyl β-D-1-thiogalactopyranoside (IPTG, BLDpharm, Shanghai, China). Protein expression was performed at 20 °C for 16 h. Cell pellets corresponding to disordered protein fragments (p53^1−94^, p53^TAD1^, and p53^TAD2^) were harvested and sonicated in denaturating lysis buffer (50 mM Tris-HCl pH 7.5, 150 mM NaCl, 20 mM Imidazole, 6 M urea), and cell pellets corresponding to folded fragments (p53^DBD^ and p53^1−312^) were harvested and sonicated in non-denaturating lysis buffer (50 mM Tris-HCl pH 7.5, 150 mM NaCl, 20 mM Imidazole, 2 mM tris(2-carboxyethyl)phosphine(TCEP, BLDpharm, Shanghai, China). After sonication, samples were centrifuged at 6198 rcf for 45 min at 4 °C. ZZ-His_6_ tag proteins were purified from the lysate using Ni-NTA agarose beads (Qiagen, Hilden, Germany), and then ZZ-His_6_ tag was cleaved by the addition of 2 (*w*/*w*) % His_6_-tagged TEV protease for 16 h at 4 °C. Desalting of proteins into low imidazole buffer (50 mM Tris-HCl pH 7.5, 150 mM NaCl, 2 mM Imidazol) was performed using a desalting column (HiPrep 26/10, GE Healthcare, Chicago, IL, USA) on an ÄKTA Pure system (GE Healthcare, Chicago, IL, USA). Untagged proteins were separated from uncleaved protein by performing a second affinity purification using Ni-NTA beads. Lastly, size exclusion chromatography purification was performed in the buffer of interest using a gel filtration column (Superdex 75 Increase, GE Healthcare (Chicago, IL, USA) for p53^1−94^, p53^DBD^, p53^1−312^; and Superdex Peptide, GE Healthcare (Chicago, IL, USA) for p53^TAD1^ and p53^TAD2^). Their absorbance at 280 nm was used to estimate protein concentrations by assuming that the ε at 280 nm was equal to the theoretical ε value.

### 4.3. NMR Spectroscopy

#### 4.3.1. Binding Assays

Proteins were purified into 20 mM Hepes (Carl Roth, Karlsruhe, Germany) pH 7.0, 50 mM NaCl (Carl Roth, Karlsruhe, Germany), 2 mM TCEP, and 0.04% (*w*/*v*) NaN_3_. All binding experiments were performed at 25 °C on a Bruker Avance Neo 600 MHz spectrometer (Bruker, Rheinstetten, Germany) equipped with a triple-resonance probehead. ^1^H-^15^N HSQC spectra (hsqcetfpf3gpsi, 16 scans, 128 points in F1, 1024 points in F2) were recorded for the titrations of 50 µM ^15^N-p53 ^1−312^, ^15^N-p53^DBD^, ^15^N-p53^TAD1^, ^15^N- p53^1−94^, and ^15^N-p53^TAD2^ with one stoichiometric equivalent of either PR25 or GR25. Reference ^1^H-^15^N HSQC spectra were recorded for ^15^N-p53 ^1−312^, ^15^N-p53^DBD^, ^15^N-p53^TAD1^, ^15^N- p53^1−94^, and ^15^N-p53^TAD2^ without a binding partner. Data analysis was performed using Bruker Topspin 4.02 (Bruker, Rheinstetten, Germany) and ccpnmr (version 2.5.) [70]. CSPs of p53^1−94^ and p53^TAD2 1^H-^15^N cross-peaks upon binding to PR25/GR25 were calculated using the following equation:(1)$$\mathrm{CSP}=\sqrt{{\left(\delta_{H}\right)}^{2}+\frac{{\left(\delta_{N}\right)}^{2}}{10}}$$

#### 4.3.2. Resonance Assignment and NOE Experiments

The following 3D spectra were acquired for resonance assignment of the 300 µM ^13^C,^15^N-labeled p53^1−94^ and the 300 µM ^13^C,^15^N-labeled p53^1−94^ in complex with 400 µM unlabeled PR25 and GR25: noesyhsqcgpwgx13d with 200 ms NOESY mixing time experiment for ^13^C/^15^N filtered, ^13^C-edited NOESY-HSQC (16 scans, 200 points in F1, 40 points in F2, 1024 points in F3), CBCA(CO)NH (cbcaconhgp3d with 8 scans, 110 points in F1, 58 points in F2, 1024 points in F3 for unbound and PR25 complex; 8 scans, 200 points in F1, 66 points in F2, 1024 points in F3 for GR25 complex). Data analysis was performed using Bruker Topspin 4.02 and ccpnmr (version 2.5) [70].

The ^15^N{^1^H} heteronuclear NOE experiments (hsqcnoef3gpsi, 16 scans, 256 points in F1, 1024 points in F2) were recorded for unbound p53^1−94,^ and p53^1−94^ in the presence of either PR25 or GR25 with a saturation period/total interscan delay of 3.0 s. HetNOE values were calculated by division of intensity of saturated spectra to intensity of unsaturated spectra. Standard deviations of saturated and unsaturated spectra were calculated using 10 additional random peaks. Error bars for heteronuclear NOE values were derived from error propagation calculation using a standard deviation of 10 arbitrarily chosen noise peaks in saturated and unsaturated spectra.

Random coil chemical shifts were used as reference values to calculate secondary chemical shift based on ncIDP (neighbor corrected IDP library) using the following formula [71,72,73]:(2)$$\Delta \delta ={\;\delta }_{\mathrm{observed}}-\delta_{\mathrm{random}\;\mathrm{coil}}$$

### 4.4. Fluorescence Polarization Measurements

N-terminally fluorescein isothiocyanate (FITC)-labelled PR25/GR25 were dissolved into 20 mM Hepes pH 7.0, 50 mM NaCl, 2 mM TCEP, and 0.04% (*w*/*v*) NaN_3_. Measurements were taken at room temperature in black 384-well plates using a ClarioStar Plus (BMG labtech, Ortenberg, Germany) spectrophotometer. Filters were selected as a function of FITC optical characteristics (λ_ex_= 495 nm, and λ_em_= 530 nm). Then, 100 nM FITC-labelled PR25/GR25 were incubated with increasing concentrations of purified p53^1−94^ and p53^TAD2^ in a final volume of 35 µL. The polarization data were fitted using graphpad prism 8 with the following equation:(3)$$P=P_{0}+\frac{P_{\max}\times L}{L\;+{\;K}_{d}}$$

Here, P_0_ represents the polarization of FITC-labelled peptides in absence of p53^1−94^ and p53^TAD2^, and P_max_ to the highest polarization of the binding curve corresponding to the saturation of the interaction. L corresponds to the concentration of p53^1–94^ and p53^TAD2^ proteins, and K_d_ is the dissociation constant.

### 4.5. Turbidity Assay

All proteins (p53^1−94^, p53^TAD2^, p53^DBD^, and p53^1−312^) and PR25/GR25 were prepared in 20 mM Hepes pH 7.0, 50 mM NaCl, 2 mM TCEP, and 0.04% (*w*/*v*) NaN_3_. Fixed concentrations (50 µM) of proteins were incubated with increasing concentrations of PR25/GR25. Turbidity measurements were conducted at 620 nm in 96-well plates with 90 μL samples using a ClarioStar Plus (BMG Labtech, Ortenberg, Germany) spectrophotometer. Each experiment was performed in three replicates.

### 4.6. Differential Interference Contrast Microscopy

All proteins (p53^1−94^, p53^TAD2^, p53^DBD^, and p53^1−312^) and PR25/GR25 were prepared in 20 mM Hepes pH 7.0, 50 mM NaCl, 2 mM TCEP, and 0.04% (*w*/*v*) NaN_3._ The 30 μL sample was plated on a 30 mm No. 1 round glass coverslip and mounted on an Observer D1 microscope with 100×/1.45 oil immersion objective (Zeiss, Oberkochen, Germany). Protein droplets were viewed using HAL 100 halogen lamp, and images were captured with an OrcaD2 camera (Hamamatsu, Shizuoka, Japan) using VisiView 4.0.0.13 software (Visitron Systems GmbH, Puchheim, Germany). Droplet formation was induced by the addition of PR25/GR25 in a fixed concentration of protein. Images were taken every 5 min until 30 min after the addition of peptides. Microscopy images were processed using Fiji/ImageJ 1.53a software (Bethesda, US), applying linear enhancement for brightness and contrast.

## Figures and Tables

**Figure 1 ijms-22-11431-f001:**
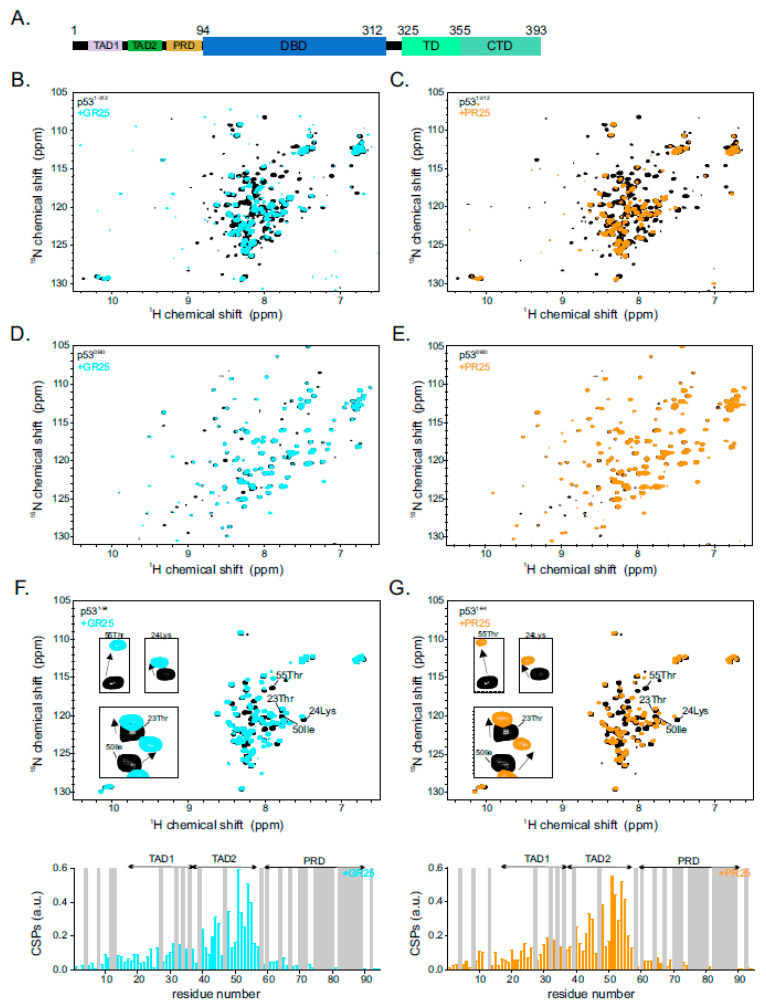
p53 directly interacts with poly-PR/GR dipeptide repeats: (**A**) p53 domain organization including transactivation domain 1 and 2 (TAD1 and TAD2), proline rich domain (PRD), DNA binding domain (DBD), tetramerization domain (TD), and C-terminal domain (CTD); (**B**,**C**) ^1^H-^15^N HSQC spectrum of ^15^N-labeled p53^1−312^ at 50 µM in the absence (black) or presence of one stoichiometric equivalent of either GR25 (cyan in **B**) or PR25 (orange in **C**); (**D**,**E**) ^1^H-^15^N HSQC spectrum of ^15^N-labeled p53^DBD^ at 50 µM in the absence (black) or presence of one stoichiometric equivalent of either GR25 (cyan in **D**) or PR25 (orange in **E**); (**F**) ^1^H-^15^N HSQC spectrum of ^15^N-labeled p53^1−94^ at 50 µM in the absence (black) or presence of one stoichiometric equivalent of GR25 (cyan; upper panel). Corresponding chemical shift perturbations (CSPs) of the ^1^H-^15^N HSQC p53^1−94^ cross-peaks are shown in a bar-plot at the bottom panel; (**G**) ^1^H-^15^N HSQC spectrum of ^15^N-labeled p53^1−94^ at 50 µM in the absence (black) and presence of one stoichiometric equivalent of PR25 (orange; upper panel). Corresponding CSPs are shown in a bar plot at the bottom panel. Unassigned residues are indicated in grey.

**Figure 2 ijms-22-11431-f002:**
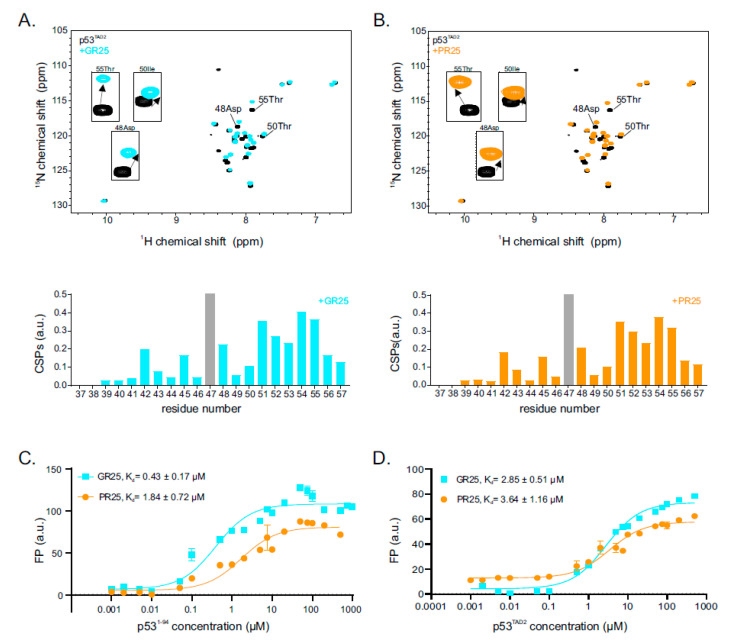
TAD2 is sufficient for binding to PR25/GR25. (**A**) ^1^H-^15^N HSQC spectrum of ^15^N-labeled p53^TAD2^ at 50 µM in the absence (black) and presence of one stoichiometric equivalent of GR25 (cyan; upper panel). Corresponding CSPs are shown in a bar-plot in the bottom panel; (**B**) ^1^H-^15^N HSQC spectrum of ^15^N-labeled p53^TAD2^ at 50 µM in the absence (black) and presence of one stoichiometric equivalent of PR25 (orange; upper panel). Corresponding CSPs are shown in a bar-plot at the bottom panel. Unassigned residues are indicated in grey; (**C**) Fluorescence polarization (FP) measurements of fluorescein isothiocyanate (FITC)-labeled GR25 (cyan) or PR25 (orange) with increasing concentrations of p53^1−94^. K_d_ values are shown and were calculated by assuming a 1:1 complex formation. Graph represents the mean of three experiments and the reported errors correspond to the SD of the fit; (**D**) FP measurements of FITC-labeled GR25 (cyan) or PR25 (orange) with increasing concentrations of p53^TAD2^. K_d_ values are shown and were calculated by assuming a 1:1 complex formation. Graph represents the mean of three experiments and the reported errors correspond to the SD of the fit.

**Figure 3 ijms-22-11431-f003:**
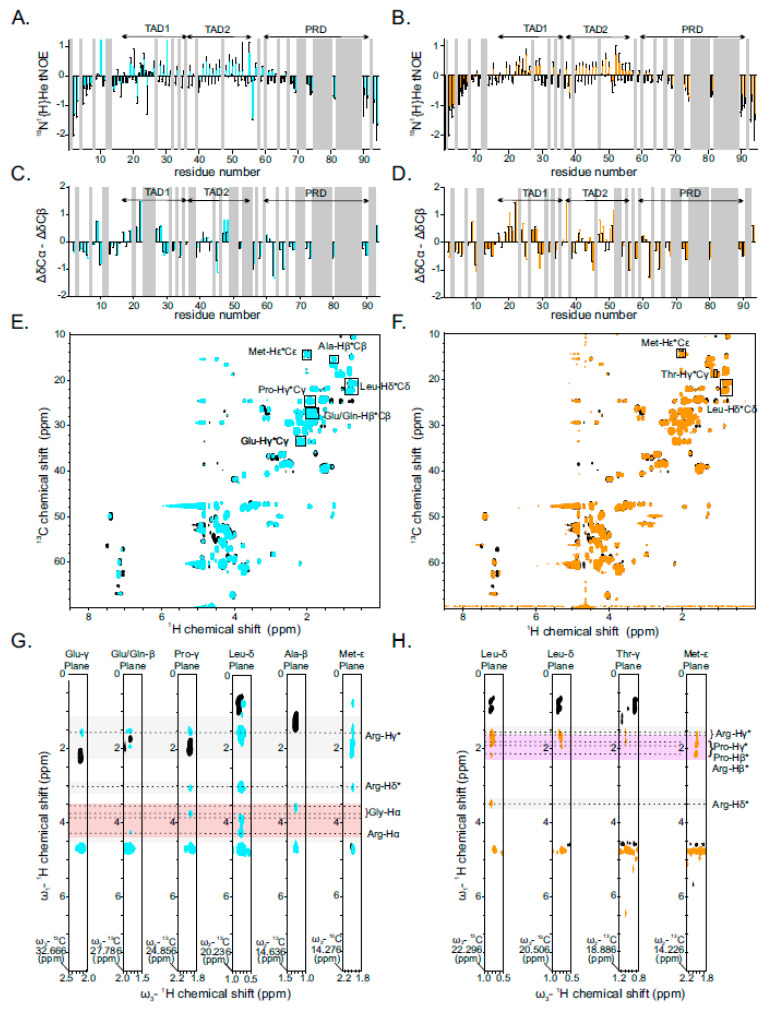
Binding of PR25/GR25 to p53^1−94^ increases its rigidity and α-helical secondary structure propensity via electrostatic and/or hydrophobic interactions. (**A**,**B**) ^15^N{^1^H} heteronuclear NOEs are plotted for p53^1−94^ residues in the absence (black) or presence of GR25 (cyan in **A**) or PR25 (orange in **B**), respectively. Error bars for heteronuclear NOE values were derived from error propagation calculation using standard deviation of 10 arbitrarily chosen noise peaks in saturated and unsaturated spectra; (**C**,**D**) Plot of difference between the secondary ^13^C chemical shifts of the Cα (noted as ΔδCα) and the Cβ (noted as ΔδCβ) nuclei of p53^1−94^ in the absence (black) or presence of GR25 (cyan in **C**) and PR25 (orange in **D**), respectively. Secondary chemical shifts were obtained by subtracting the random coil chemical shifts (predicted by ncIDP) from observed chemical shifts. Differences between the secondary chemical shift deviations (ΔδCα - ΔδCβ) were plotted against the amino acid residue numbers, taking into account next neighbor effects; (**E**,**F**) ^1^H-^13^C HSQC spectra of ^13^C-labeled p53^1−94^ in the absence (black) or presence of either GR25 (cyan in **E**) or PR25 (orange in **F**). Residues with NOE cross peaks are indicated by black squares; (**G**) Slices from ^1^H^1^H planes of a 3D ^13^C/^15^N filtered, ^13^C-edited NOESY-HSQC spectrum recorded on ^13^C,^15^N-labelled p53^1−94^ in the presence of 1.33 stoichiometric equivalents of unlabeled GR25. NOESY cross peaks are shown in cyan. Representative two-dimensional planes are shown for different ^13^C chemical shifts and illustrate intermolecular NOEs involving Glu-Hγ*Cγ, Glu/Gln-Hβ*Cβ, Pro- Hγ*Cγ, Leu-Hδ*Cδ, Ala- Hβ*Cβ, and Met-Hε*Cε (from left to right) and their corresponding signals are indicated by dashed lines; cross peaks of these residues to ^1^H chemical shifts characteristic for glycine and arginine residues are labeled. Black and cyan peaks correspond to positive and negative peaks, respectively; (**H**) Slices from ^1^H^1^H planes of a 3D ^13^C/^15^N filtered, ^13^C-edited NOESY-HSQC spectrum recorded on ^13^C,^15^N-labelled p53^1−94^ in the presence of 1.33 stoichiometric equivalents of unlabeled PR25. NOESY cross peaks are shown in orange. Representative two-dimensional planes are shown at different ^13^C chemical shifts and illustrate intermolecular NOEs involving Leu-Hδ*Cδ, Thr- Hγ*Cγ, and Met-Hε*Cε (from left to right) and their corresponding signals are indicated by dashed lines. Cross peaks of these residues to ^1^H chemical shifts characteristic for glycine and arginine residues are labeled. Black and orange peaks correspond to positive and negative peaks, respectively.

**Figure 4 ijms-22-11431-f004:**
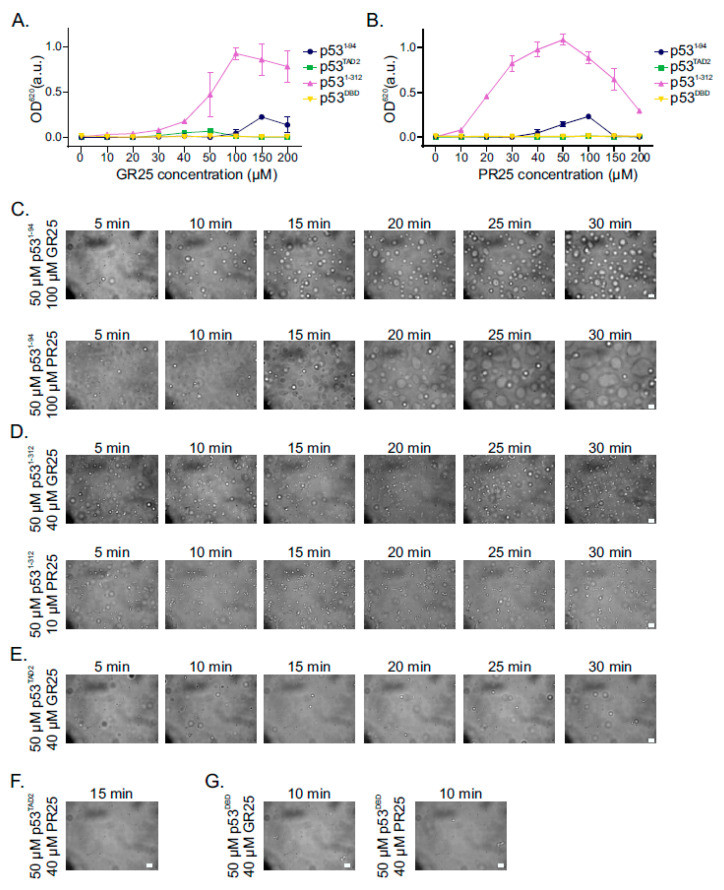
PR25/GR25 modulate LLPS of p53. (**A**,**B**) Turbidity assays to quantify phase separation of p53 domains with fixed p53 concentrations (50 μM) and increasing either GR25 concentration in (**A**) or PR25 concentration in (**B**). Values represented mean ± SD (n = 3); (**C**) DIC microscopy images of p53^1−94^ at 50 μM in the presence of either 100 μM GR25 (upper panel) or 100 μM PR25 (bottom panel); (**D**) DIC microscopy images of p53^1−312^ at 50 μM in the presence of either 40 μM GR25 (upper panel) or 10 μM PR25 (bottom panel); (**E**,**F**) DIC microscopy images of p53^TAD2^ at 50 μM in the presence of either 40 μM GR25 in (**E**) or 40 μM PR25 in (**F**); (**G**) DIC microscopy images of p53^DBD^ at 50 μM in the presence of either 40 μM GR25 (**left**) or 40 μM PR25 (**right**). Scale bar, 10 μm.

**Figure 5 ijms-22-11431-f005:**
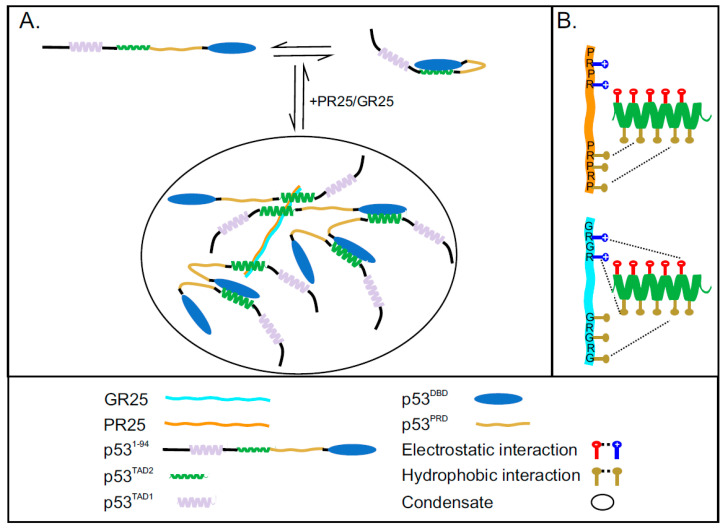
Interaction between p53 and PR25/GR25 is mediated by mainly TAD2 and phase separation of p53 is induced by PR25/GR25. (**A**) p53 is found in two forms, which are open and p53^DBD^ bound to p53^TAD2^ (closed state, auto-inhibited) [64,65,66]. The addition of PR25/GR25 induces phase separation of p53, which is mediated by a combination of intermolecular interactions between p53^DBD^ and p53^TAD2^ and intermolecular interactions between p53^TAD2^ and PR25/GR25; (**B**) Schematic representation of the network of hydrophobic and electrostatic interactions between p53^TAD2^ and PR25/GR25.

## Data Availability

NMR chemicals have been submitted to the Biological Magnetic Resonance Bank (BMRB) under the accession numbers BMRB 51125 (p53^TAD2^) and BMRB 51124 (p53^1−94^). All other data discussed in this study are included in the text.

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
