# Peer review of "p53 Transactivation Domain Mediates Binding and Phase Separation with Poly-PR/GR"

_ijms, 2021, doi:10.3390/ijms222111431_

Round 1

Reviewer 1 Report

This very nice study is a detailed characterization of the interactions between the intrinsically disordered N-terminal domain of p53 and the aberrant poly-PR/GR repeats produced by the C9orf72 gene.  The interaction is an important phenomenon involved in the progression of amyotrophic lateral sclerosis, and is of great scientific interest. The authors isolate the TAD2 domain as the main interface of this interaction, with details regarding the nature of the binding and consequences to the secondary structure of the TAD2 domain.

 The manuscript seems very sound, and I could not find anything to point out regarding the experimental design, execution and interpretation. A very minor point; in the latter portion of their Introduction, the authors allude to a contribution by the adjacent TAD1 domain during TAD2:Poly-PR/GR binding, specifically an assistive effect that enhances binding affinity.  This idea was not expanded upon in the results, I believe. Presumably the authors are referring to the differences in binding curves in Fig. 2 C and D? Maybe the authors might clarify this point a bit further in the text. 

 Another rather insignificant point, regarding Figure 4 C to F; the superscript notations in the vertical axis are particularly hard to read in the original pdf.  It would be appreciated if this could be improved somehow.

Author Response

A very minor point; in the latter portion of their Introduction, the authors allude to a contribution by the adjacent TAD1 domain during TAD2:Poly-PR/GR binding, specifically an assistive effect that enhances binding affinity.  This idea was not expanded upon in the results, I believe. Presumably the authors are referring to the differences in binding curves in Fig. 2 C and D? Maybe the authors might clarify this point a bit further in the text. 

We thank the reviewer for her/his very positive feedback to our manuscript. We added a discussion of the contribution of TAD1 for PR25/GR25 binding and affinities of p53TAD2 and p531-94 for PR25/GR25 in the revised version of the manuscript (lines 141-144).

“Compared to p53TAD2, p531-94 has approximately 2-fold/6-fold higher affinities for PR25 and GR25 binding, respectively (Figures 2C and D). Even though p53TAD1 is insufficient to bind PR25 and GR25 (SF1A and B), it enhances binding to PR25/GR25 in the p531-94 construct (Figures 2C and D).”

Another rather insignificant point, regarding Figure 4 C to F; the superscript notations in the vertical axis are particularly hard to read in the original pdf.  It would be appreciated if this could be improved somehow.

We increased the size of the text in Figure 4.

Reviewer 2 Report

The manuscript describes solid data from different experimental approaches elucidating direct interactions between p53 and PR/GR peptides, the effect of these interactions on LLPS, as wells as reasonable model (Figure 5) explainging potential molecular mechanisms involved. NMR experiments presented are well planned and thorough.

My only minor concern is about over-interpretation of the data shown in Figure 3: a/b - hetNOEs and c/d - delta Cs. The figure itself in my download was low quality, so it was hard to see the exact changes, but still they were minimal, and although some of the negative hNOE in TAD1/TAD2 regions seem to disappear, some of the positive ones were reduced in value as well. So am not so sure that the statement on page 6 is exactly accurate ("We observed that HetNOEs increased in the presence of PR25/GR25, in the regions comprising TAD1 and TAD2 (Figures 3A and B), which indicates that binding of PR25/GR25 enhances the rigidity of these p53 domains"). I agree that rigidity might increased upon binding, but the prove is not a clear cut. Same is true for chemical shifts differences (3C/D) - only a couple of residues in TAD2  unambiguously shows increased propensity to form helix (and i think those should be named, as it is really hard to see the numbers in the figure). 

Other minor concern is low quality of  Figure 4A/B - it was really hard to read what construct corresponds to a particular symbol on the curve and p53DBD symbols are not seen at all.

Author Response

My only minor concern is about over-interpretation of the data shown in Figure 3: a/b - hetNOEs and c/d - delta Cs. The figure itself in my download was low quality, so it was hard to see the exact changes, but still they were minimal, and although some of the negative hNOE in TAD1/TAD2 regions seem to disappear, some of the positive ones were reduced in value as well. So am not so sure that the statement on page 6 is exactly accurate ("We observed that HetNOEs increased in the presence of PR25/GR25, in the regions comprising TAD1 and TAD2 (Figures 3A and B), which indicates that binding of PR25/GR25 enhances the rigidity of these p53 domains"). I agree that rigidity might increased upon binding, but the prove is not a clear cut. Same is true for chemical shifts differences (3C/D) - only a couple of residues in TAD2 unambiguously shows increased propensity to form helix (and i think those should be named, as it is really hard to see the numbers in the figure). 

We thank the reviewer for her/his very positive feedback to our manuscript. Residues, which seem to disappear (residue #32 and #39), do not have assignment in the unbound form. We apologize for the confusion. We have added gray bars to outline this in the revised version of the manuscript. In the revised version, residues, which are becoming more flexible upon binding of PR25/GR25, are mentioned in the text and discussed in detail (lines 177-180). Residues in the TAD2 region, which show increased α-helical secondary structure propensity upon PR25/GR25 binding, are named in the text as suggested (lines 193-194).

Other minor concern is low quality of Figure 4A/B - it was really hard to read what construct corresponds to a particular symbol on the curve and p53DBD symbols are not seen at all.

We thank the reviewer for her/his suggestion. We increased the size of the text in Figure 4. The color of p53DBD in Figures 4A and B were changed to improve visibility.